# Age-Related Alterations in the Level and Metabolism of Serotonin in the Brain of Males and Females of Annual Turquoise Killifish (*Nothobranchius furzeri*)

**DOI:** 10.3390/ijms24043185

**Published:** 2023-02-06

**Authors:** Valentina S. Evsiukova, Alla B. Arefieva, Ivan E. Sorokin, Alexander V. Kulikov

**Affiliations:** 1Department of Psychoneuropharmacology, Federal Research Center Institute of Cytology and Genetics, Siberian Branch of the Russian Academy of Sciences, 630090 Novosibirsk, Russia; 2Department of Genetic Collections of Neural Disorders, Federal Research Center Institute of Cytology and Genetics, Siberian Branch of the Russian Academy of Sciences, 630090 Novosibirsk, Russia; 3Department of Monogenic Forms of Human Common Disorders, Federal Research Center Institute of Cytology and Genetics, Siberian Branch of the Russian Academy of Sciences, 630090 Novosibirsk, Russia

**Keywords:** turquoise killifish, aging, serotonin, tryptophan hydroxylase, monoamine oxidase, brain

## Abstract

The annual turquoise killifish (*Nothobranchius furzeri*) is a laboratory model organism for neuroscience of aging. In the present study, we investigated for the first time the levels of serotonin and its main metabolite, 5-hydroxyindoleacetic acid, as well as the activities of the key enzymes of its synthesis, tryptophan hydroxylases, and degradation, monoamine oxidase, in the brains of 2-, 4- and 7-month-old male and female *N. furzeri*. The marked effect of age on the body mass and the level of serotonin, as well as the activities of tryptophan hydroxylases and monoamine oxidase in the brain of killifish were revealed. The level of serotonin decreased in the brain of 7-month-old males and females compared with 2-month-old ones. A significant decrease in the tryptophan hydroxylase activity and an increase in the monoamine oxidase activity in the brain of 7-month-old females compared to 2-month-old females was shown. These findings agree with the age-related alterations in expression of the genes encoding tryptophan hydroxylases and monoamine oxidase. *N. furzeri* is a suitable model with which to study the fundamental problems of age-related changes of the serotonin system in the brain.

## 1. Introduction

The brain serotonin (5-HT) system plays a key role in the regulation of neuronal plasticity [1], numerous physiological functions and various kinds of behavior [2], while its dysfunctions are associated with grave psychopathologies [3,4,5,6,7]. The key enzyme of 5-HT synthesis in the mammalian brain is tryptophan hydroxylase 2 (TPH2) hydroxylating L-tryptophan to 5-hydroxytryptophan [8,9]. Indeed, the genetic [10,11,12,13,14] or pharmacological [15,16,17] deficiency of TPH2 activity dramatically decreases the level of 5-HT in the mouse brain. The synthesized 5-HT released from the endings of 5-HT neurons into the synaptic cleft regulates multiple physiological functions and the kinds of behavior via 14 different kinds of 5-HT receptors [18,19,20]. The duration of physiological and behavioral effects of 5-HT are regulated by the transmembrane 5-HT transporter (SERT) that reuptakes 5-HT from the synaptic cleft into the 5-HT neurons [21], where the enzymes monoamine oxidases A and B (MAOA and MAOB) oxidize 5-HT to 5-hydroxyindoleacetic acid (5-HIAA) [22,23,24]. Inhibitors of SERT and MAOs increase the 5-HT level in the synaptic cleft and are clinically effective antidepressants [3,4,5,6,7]. In general, the levels of 5-HT, 5-HIAA, the 5-HT turnover rate (5-HIAA/5-HT), TPH2, MAOA and MAOB activities indicate the functional activity of the brain 5-HT system [3,4,5,6,7].

Age-related alterations in the functional activity of the brain 5-HT system frequently accompany psychic disorders in senior patients [25]. The vast amount of knowledge concerning age-related changes in the brain biogenic amines was based on results obtained from senior people using the positron emission tomography assay and postmortem studies as well as experiments carried out on old laboratory monkeys and rodents [see reviews [26,27,28]]. However, these studies are dramatically limited due to ethical norms and the relatively long life span of laboratory monkeys (>20 years) and rodents (>3 years).

Annual turquoise killifish, *Nothobranchius furzeri*, inhabit ephemeral ponds in southeastern Africa. Their fertilized eggs survive the dry season in diapause. The larvae hatch immediately after the pond is filled with water, grow rapidly, reach sexual maturity within 4–6 weeks, continuously mate and spawn during the wet season and die at the age of 6–8 months [29,30,31]. Six- to eight-month-old *N. furzeri* show morphological and behavioral hallmarks of aging, such as reduced coloration in males, emaciation, spinal curvature, spine and face malformations [29,31], reduced spontaneous, exploratory activities [32] and impaired learning performance [33,34]. *N. furzeri* is a suitable model organism for the study of human aging and age-related disorders [35,36,37,38,39].

The brain 5-HT system regulates the physiological functions and behavior of killifish. Indeed, chronic treatment with small doses of a 5-HT transporter blocker, fluoxetine, decreases body size as well as increases fecundity and sociability in *N. furzeri* [40,41]. The anatomical, cellular and neurochemical organization of the 5-HT system in fish is homologous to those of mammals [42,43]. However, there are some differences in the molecular organization of the 5-HT system in the brain of *N. furzeri*. Namely, three different genes, *Tph1a*, *Tph1b* and *Tph2*, encoding three different tryptophan hydroxylases (TPH1a, TPH1b, TPH2), as well as the only *Mao* gene encoding the only monoamine oxidase (MAO), are expressed in the brain of *N. furzeri* [44]. Recently dramatic age-related alterations in the expression of the main 5-HT related genes encoding TPH1b, TPH2 and MAO in the brain of *N. furzeri* were shown [44]. At the same time, the levels of the 5-HT, 5-HIAA levels, 5-HIAA/5-HT turnover rate and TPHs and MAO activities indicate the functional activity of the brain 5-HT system on the molecular level. So far only one study concerning the levels of 5-HT and 5-HIAA in the brain of mature (4-month-old) males and females of *N. furzeri* has been published [see [45]].

Therefore, the aim of the present study was to investigate age-dependent alterations in the levels of 5-HT, 5-HIAA and 5-HIAA/5-HT turnover rate and TPHs and MAO activities in the brain of young, mature and old males and females of *N. furzeri*.

## 2. Results

### 2.1. Body Mass in 2-, 4- and 7-Month-Old Males and Females of N. furzeri

Marked effects of the “Age” (F(2,42) = 19.86, *p* < 0.001), “Sex” (F(1,42) = 140.25, *p* < 0.001) factors and the factor interaction (F(2,42) = 8.23, *p* < 0.001) on body mass of *N. furzeri* were revealed. As expected, in each age group the males were heavier than the females (Figure 1). The body mass of males progressively increased with age, while the body mass of females did not alter with age (Figure 1).

### 2.2. Levels of 5-HT, 5-HIAA and 5-HIAA/5-HT Rate in the Brain of 2-, 4- and 7-Month-Old Males and Females of N. furzeri

A significant effect of the “Age” factor on the 5-HT level, the 5-HIAA/5-HT turnover rate, but not on the 5-HIAA level in the brain of *N. furzeri* was shown (Table 1). The level of 5-HT progressively decreased, while the 5-HIAA/5-HT turnover rate progressively increased with age in the brain of males and females of *N. furzeri* (Figure 2). The 5-HIAA level did not change with age in the brain of males and females of *N. furzeri* (Figure 2). At the same time, no effect of the “Sex” factor and the factor interaction on these traits was observed (Table 1). In each age group, the 5-HT and 5-HIAA levels and the 5-HIAA/5-HT turnover rate in males did not differ from those in females (Figure 2).

### 2.3. Activity of TPH in the Brain of 2-, 4- and 7-Month-Old Males and Females of N. furzeri

A significant effect of the “Age” factor (F(2,42) = 3.48, *p* = 0.04), but not of the “Sex” factor (F(2,42) = 1.63, *p* = 0.21), and the factor interaction (F(2,42) = 2.42, *p* = 0.1) on TPH activity in the brain of *N. furzeri* was found. The TPH activity decreased in the brain of mature and old females compared to young females (Figure 3).

### 2.4. Activity of MAO in the Brain of 2-, 4- and 7-Month-Old Males and Females of N. furzeri

Marked effect of the “Age” factor (F(2,42) = 6.24, *p* = 0.004), but not of the “Sex” factor (F(2,42) < 1), and the factor interaction (F(2,42) < 1) on the MAO activity in the brain of *N. furzeri* was found. The MAO activity increased in the brain of old females compared to young and mature females (Figure 4).

### 2.5. Correlations between the Levels of the 5-HT, 5-HIAA and 5-HIAA/5-HT Turnover Rate and TPHs and MAO Activities in the Brain of Male and Female N. furzeri

Significant positive correlations between the TPH activity, the 5-HT and 5-HIAA levels in the brain of male and female *N. furzeri* were shown (Table 2). As expected, the 5-HIAA/5-HT turnover rate positively correlated with the 5-HIAA level and negatively correlated with the 5-HT level (Table 2). At the same time, no statistically significant correlation between the MAO activity on one hand and the 5-HT and 5-HIAA levels and the 5-HIAA/5-HT turnover rate on the other hand in the brain of males and females of *N. furzeri* was found (Table 2).

Principal component analysis revealed three factors that defined 93.1% variability in the 5-HT and 5-HIAA levels, the 5-HIAA/5-HT turnover rate and the TPHs and MAO activities in brains of males and females. The first factor negatively correlates with the TPH activity, the 5-HT level and the 5-HIAA/5-HT turnover rate (Table 3, Figure 5). It reflects a decrease in the TPH activity and the 5-HT level and an increase in the 5-HIAA/5-HT turnover rate in old killifish. The second factor positively correlates with the TPH activity and the 5-HIAA level (Table 3, Figure 5). Finally, the third factor positively correlates only with the MAO activity (Table 3). This three-factor model reveals a significant effect of age on these traits in males and females (F(6,84) = 9.43, *p* < 0.0001).

Age-dependent alterations in the first (F(2,42) = 30.23, *p* < 0.0001), but not in the second (F(2,42) < 1) and the third (F(2,42) = 3.12, *p* = 0.055) factors, were revealed. Statistically significant differences between 2- and 4- (*p* = 0.0002), 2- and 7- (*p* < 0.0001) and 4- and 7- (*p* = 0.0057) month-old males were revealed (Figure 5).

## 3. Discussion

In this study, three age groups of killifish, namely (1) relatively young (2-month-old), (2) mature (4-month-old) and (3) old (7-month-old), when natural death of fish begins to occur [29,30,31], were investigated. The results confirmed and verified earlier findings [44,45] that: (1) males are heavier than females of the same age; (2) the body mass of males progressively increased with age, while no age-related alteration in body masses in females was seen; and (3) sex does not affect the 5-HT and 5-HIAA levels or the 5-HIAA/5-HT turnover rate in the brain.

Some age-related alterations in the expression of key 5-HT-related genes in the brain of male and female *N. furzeri* kept in large groups in the tanks of 125 L were shown previously [44]. The main aim of the present study was to investigate the age-related alterations in the 5-HT and 5-HIAA levels and the 5-HIAA/5-HT turnover rate as well as in the TPH and MAO activities in the brain of males and females of *N. furzeri*. In this study, killifish kept in small groups (one male and one female) in tanks of 6.8 L were used. It was shown that the tank size did not affect the 5-HT and 5-HIAA levels in the brain of males and females of *N. furzeri* [45].

The progressive decrease in the 5-HT level and the increase in the 5-HIAA/5-HT turnover rate in the brain of males and females with age were revealed. The statistically significant decrease in TPH activity and the increase in MAO activity in the brain of 7-month-old (old) females compared to 2-month-old (young) females were also shown. Similar (but statistically insignificant) age-related alterations in TPH and MAO activities were observed as tendencies in the brain of males.

The age-related changes in TPH activity seems to be a cause of the observed age-related alterations in the 5-HT and 5-HIAA levels. Indeed, two later indices positively correlate with the alterations in the TPH activity in the brain of males and females. At the same time, the marked age-dependent changes in the MAO activity do not correlate with the age-related alterations in the 5-HT and 5-HIAA levels.

A probable interpretation of these finding is the following: Firstly, TPH activity is the key factor defining the 5-HT and 5-HIAA levels. The deficit of the TPH activity results in a dramatic decrease in the 5-HT and 5-HIAA levels in the brain of mice [10,11,12,13,14,15,16,17]. An irreversible inhibitor of TPH, p-chlorophenylalanine, dramatically decreases the 5-HT and 5-HIAA levels in the brain of zebrafish [46,47]. Secondly, an MAO inhibitor, pargyline, dramatically increases the 5-HT level and decreases the 5-HIAA concentration in the brain of zebrafish [46,47]. Therefore, the age-related decrease in TPH activity and increase in MAO activity together accelerate the observed age-dependent decrease in the 5-HT levels in killifish brain. At the same time, these opposite age-related dynamics of TPH and MAO activities seems to prevent any age-related changes in the 5-HIAA level in the brain of *N. furzeri*.

The observed dramatic increase in MAO in the brain of old killifish compared to young animals is in good agreement with the age-related increase in the *Mao* gene expression [44]. The age-related elevation in the *Mao* gene expression seems to cause the observed age-dependent increase in MAO activity in the brain of killifish. This result agrees well with the elevation of the MAOA activity in the brain of rats [48] and humans [49,50,51] during aging. Age-dependent TPH2 alterations in the mammalian brain are obscure. Some authors did not find any difference in the *Tph2* mRNA level in the brain structures between young and old rats [52], while other authors reported increase and decrease in the TPH2 activity in midbrain and medulla, respectively, in old rats compared to middle-aged ones [53]. In the present study, we also showed the progressive decrease in the TPH activity in the killifish brain with aging. The molecular mechanism of the age-related decrease in the TPH activity is unclear, since three genes, *Tph1a*, *Tph1b* and *Tph2* encoding corresponding enzyme molecules, are expressed in the brain of killifish [44] and the impacts of TPH1a, TPH1b and TPH2 on total enzyme activity is still obscure. Moreover, the observed age-dependent decrease in the TPH activity disagrees with absence of age-dependent dynamics in these genes expression the brain of killifish females [44].

## 4. Materials and Methods

### 4.1. Animals

The experiments were carried out on 24 males and 24 females of *N. furzeri* of the ZMZ1001 strain. The progenitors of these killifish were received from the European Research Institute of Biology of Ageing (ERIBA, Groningen, Netherlands) [44,45]. The breeding and hatching of these fish were carried out in the fish facility of Institute of Cytology and Genetics SB RAS (Novosibirsk, Russia) (supported by the budget project No. FWNR-2022-0023) according to the published protocol [54]. The larvae were fed ad libitum three times per day with freshly hatched brine shrimps (*Artemia salina*) for the first three weeks after hatching. At the age of 21 days post-hatch, young fish were reared in pairs (one male and one female) in 8 L glass tanks (20 cm in length, 20 cm in depth, 20 cm in width, water depth was 17 cm, the final volume was 6.8 L). Fresh water (pH~7.4, [Ca^++^]~50 mg/L, [Mg^++^]~17 mg/L) in the tanks was constantly filtered and aerated with a filter, model 019 (Barbus, Xiaolan, China), and its temperature was 27 °C. Every tank was equipped with a plastic plant 7 cm and a dish (10 cm in diameter, 3 cm in height) filled with sand for spawning. Every day (at 17:00), the tanks were cleaned and 20% of water was substituted by tap water filtered through the expert hard filter (Barrier, Noginsk, Russia). The 12 h light/12 h dark (“light on” mode at 09:00) photoperiod was maintained. Killifish were fed ad libitum two times per day with frozen blood worms (*Chironomus plumosus*). To avoid competition for food, we added as many worms as fish could eat in an hour. The fish were kept in these tanks until euthanasia.

There were three age groups of killifish: 62-day-old ones (8 males and 8 females), 120-day-old ones (8 males and 8 females) and 210-day-old ones (8 males and 8 females). Fish in the last age group showed all morphological markers of aging, such as reduced coloration in males, emaciation, spinal curvature, spine and face malformations. Moreover, the death frequency in this group was high. The fish were euthanized by immersion into 0.1% tricaine methanesulfonate (Sigma-Aldrich, St. Louis, MO, USA) solution and then into cold water (+2 °C); their bodies were immediately dried with dry napkins and their masses were measured using an Ohaus PA-512 electronic balance (Ohaus Corporation, Parsippany, NJ, USA) with an accuracy of 10 mg. Then, their whole brains were immediately removed, frozen with liquid nitrogen and stored at −80 °C [44,45].

In the present study the 5-HT and 5-HIAA levels, as well as TPH and MAO activities were assayed in each brain. The brain was homogenized in 150 μL of 50 mM Tris HCl, pH 7.6 with 1 mM dithiothreitol (Sigma-Aldrich, St. Louis, MO, USA) using a motor-driven grinder (Z359971, Sigma-Aldrich, St. Louis, MO, USA). An aliquot of 50 μL of the homogenate was immediately mixed with 150 μL of 0.6 M HClO_4_ and spun for 15 min at 12,700 rpm (+4 °C). The clear supernatant was diluted twice with pure water and used to assay 5-HT and 5-HIAA levels by HPLC, while the pellet was diluted in 1 mL of 0.1 M NaCl and used for protein quantitation by Bradford (Bio Rad, Hercules, CA, USA) according to the manufacturer’s protocol. The remaining 100 μL of the homogenate was spun for 15 min at 12,700 rpm (+4 °C). The clear supernatant was transferred into clear tube. The supernatant and pellet were kept at −80 °C and used to assay TPH and MAO activities, respectively.

### 4.2. Assay of 5-HT and 5-HIAA Concentrations

The 5-HT and 5-HIAA contents were assayed in the diluted supernatant by HPLC on Luna C18(2) column (5 μm particle size, L × I.D. 100 × 4.6 mm, Phenomenex, Torrance, CA, USA) with electrochemical detection (750 mV, DECADE II™ Electrochemical Detector; Antec, Hoorn, The Netherlands) as was described in a previous study [45]. The standard mixes containing 0.5, 1 and 2 ng of 5-HT and 5-HIAA (Sigma-Aldrich, St. Louis, MO, USA) were repeatedly assayed throughout the entire procedure and used to plot the calibration curves for each substance. The areas of peaks were estimated using LabSolution LG/GC software version 5.54 (Shimadzu Corporation, Kyoto, Japan) and calibrated against the calibrated curves for corresponding standards [45]. The 5-HT and 5-HIAA content was expressed in ng/mg protein assayed by Bradford.

### 4.3. Assay of TPH Activity

An aliquot of 15 μL of pure supernatant was incubated for 15 min at 27 °C in the presence of L-tryptophan (Sigma-Aldrich, St. Louis, MO, USA) (0.4 mM), cofactor 6-methyl-5,6,7,8-tetrahydropteridine (Sigma-Aldrich, St. Louis, MO, USA) (0.3 mM), decarboxylase inhibitor m-hydroxybenzylhydrazine (Sigma-Aldrich, St. Louis, MO, USA), catalase (Sigma-Aldrich, St. Louis, MO, USA) (5 μ) and 1 mM dithiothreitol in a final volume of 25 μL. The reaction was stopped with 75 μL 0.6 M HClO_4_, centrifuged for 15 min at 12,700 rpm. The clear supernatant was diluted by twofold with pure water and the 5-HTP concentration was determined in the diluted supernatant using high performance liquid chromatography (see Section 4.2) using standards of 25, 50 and 100 pmoles of 5-HTP (Sigma, USA). Another aliquot of 10 μL of supernatant was mixed with 90 μL of 0.1 M NaOH for protein quantitation by Bradford (Bio Rad, Hercules, CA, USA) according to the manufacturer’s protocol. The TPH activity was expressed in the pmoles of 5-HTP formed per minute per mg of protein measured according to Bradford.

### 4.4. Assay of MAO Activity

The MAO activity was assayed using 5-HT as a natural substrate and was defined as the amount of the product, 5-hydroxyindoleacetic aldehyde, synthesized per minute per mg of protein, according to [55] with modifications.

Briefly, the pellet obtained during TPH assay (see Section 4.3) was suspended in 100 μL of 50 mM Tris HCl, pH 7.6, using a motor-driven grinder (Z359971, Sigma-Aldrich, St. Louis, MO, USA), and then it was spun for 15 min at 500 rpm (+4 °C). The MAO activity was assayed in the turbid supernatant. An aliquot of 10 μL of the turbid supernatant was incubated for 10 min at 27 °C with 0.1 mM of 5-HT in the final volume of 25 μL. The reaction was terminated by 75 μL of 0.6 M HClO_4_, spun for 15 min at 12,700 rpm. The clear supernatant was diluted by twofold with pure water and the 5-hydroxyindoleacetic aldehyde concentration was determined in the diluted supernatant using high performance liquid chromatography (see Section 4.2) using standards of 500, 1000 and 2000 pmoles of 5-hydroxyindoleacetic aldehyde (Cymit Quimica S.L., Barcelona, Spain). Another aliquot of 10 μL of the turbid supernatant was mixed with 90 μL of 0.1 M NaOH for protein quantitation by Bradford (Bio Rad, Hercules, CA, USA), according to the manufacturer’s protocol. The MAO activity was expressed in pmoles of 5-hydroxyindoleacetic aldehyde formed per minute per mg of protein measured according to Bradford.

### 4.5. Statistics

All data were tested using the Kolmogorov’s test and met the assumption of normality. Data were presented as the mean ± SEM and analyzed by two-way ANOVA with “Age” and “Sex”, including their interaction, as independent factors. Post hoc analyses were carried out using Fisher’s LSD multiple comparison test when appropriate. The number of variables was reduced through principal component analysis with factor varimax normalized rotation. Factor loadings were calculated as the correlation coefficients between each original variable and this factor. The factor differences between the age groups were analyzed by discriminant function analysis. Statistical significance was set at *p* < 0.05. The significance of factor loadings and Pearson’s correlation coefficients was corrected according to Bonferroni.

## 5. Conclusions

The present work is a logical development of our early study concerning the age-dependent alterations in the 5-HT-related gene expression in the brain of *N. furzeri* [44]. Here, the age-related changes in the 5-HT and 5-HIAA levels, the 5-HIAA/5-HT turnover rate and TPH and MAO activities in the brains of killifish (*N. furzeri*) were studied. The following results were obtained:No effect of sex on the 5-HT and 5-HIAA levels or the TPH and MAO activities in the brain was shown.The 5-HT level was decreased, while 5-HIAA/5-HT was increased in the brain of males and females during aging.No age-related dynamic in the 5-HIAA level in the brain of males and females was observed.The TPH activity was decreased, while the MAO activity was increased in the brain of females during aging.

MAO inhibitors are promising therapeutic agents for treatment of certain age-related neurodegenerative diseases, including Parkinson’s and Alzheimer’s diseases [56,57]. *N. furzeri* is a promising model species accelerating the study of the MAO involvement in the mechanisms and therapy of age-related psychopathologies.

## Figures and Tables

**Figure 1 ijms-24-03185-f001:**
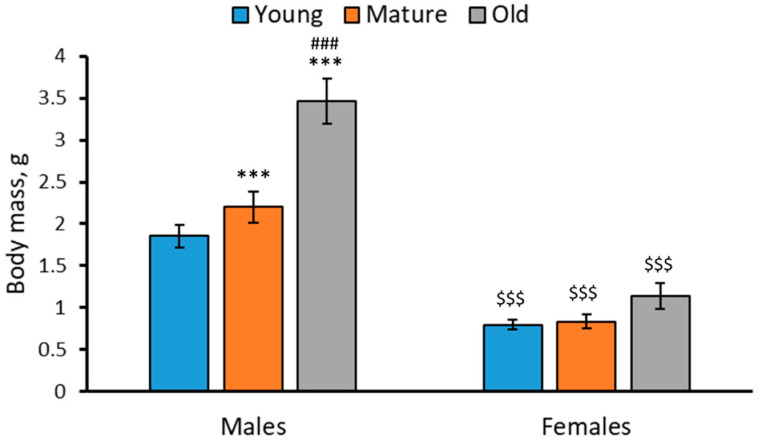
Body mass (g) in 2 (young)-, 4 (mature)- and 7 (old)-month-old males and females of *N. furzeri*. *** *p* < 0.001 vs. young males; ^###^
*p* < 0.001 vs. mature males; ^$$$^
*p* < 0.001 vs. males of the same age.

**Figure 2 ijms-24-03185-f002:**
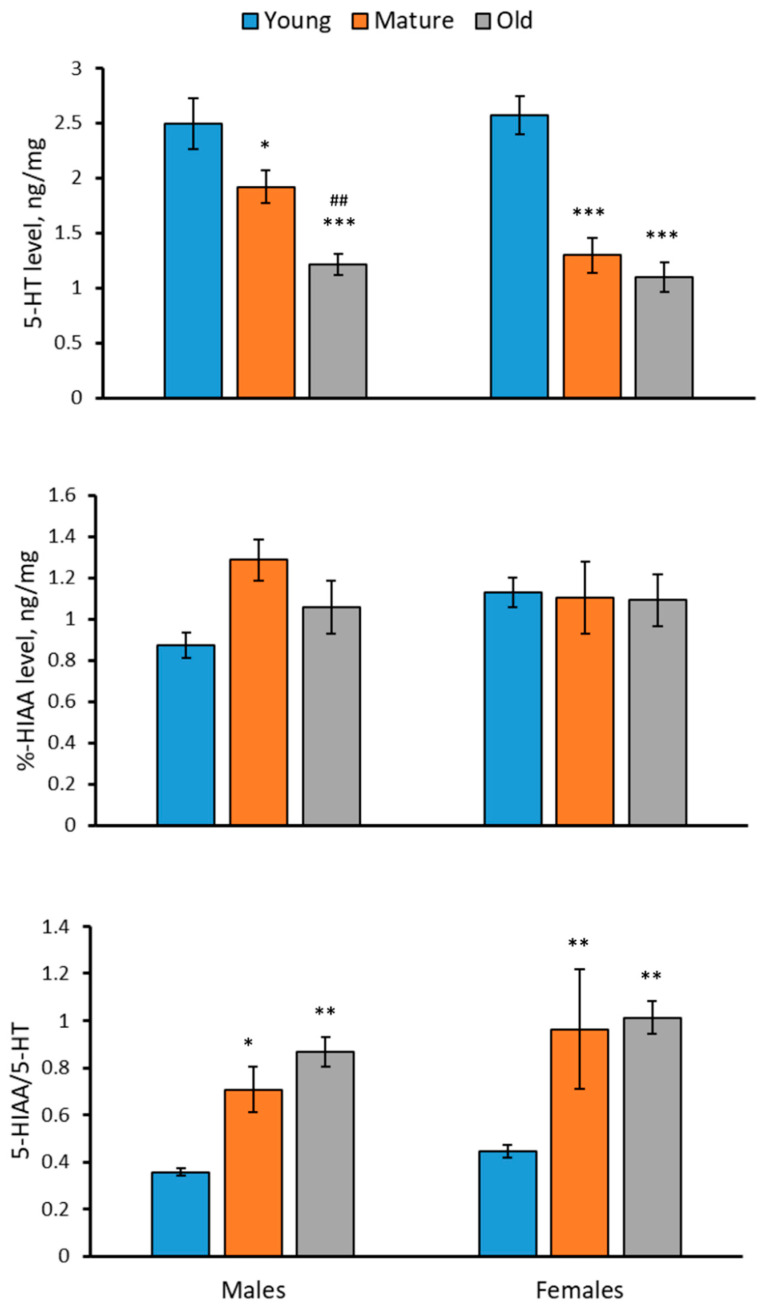
Levels of the 5-HT, 5-HIAA and 5-HIAA/5-HT turnover rate in the brain of 2 (young)-, 4 (mature)- and 7 (old)-month-old male and female *N. furzeri*. * *p* < 0.05, ** *p* < 0.01, *** *p* < 0.001 vs. young fish of the same sex; ^##^
*p* < 0.01 vs. mature males.

**Figure 3 ijms-24-03185-f003:**
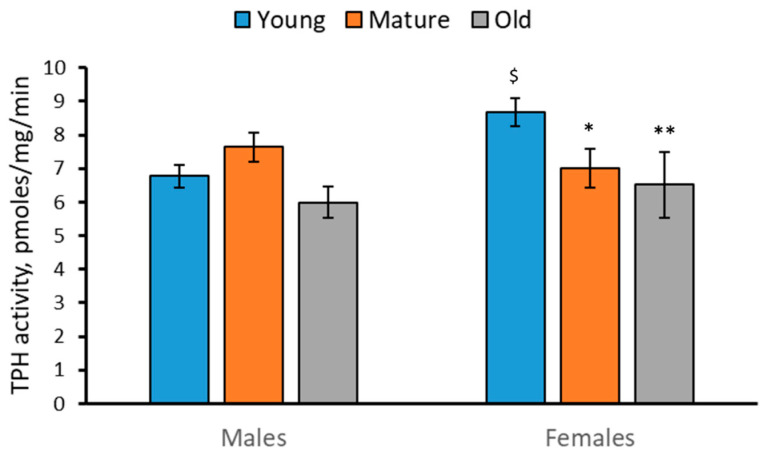
TPH activity in the brain of 2 (young)-, 4 (mature)- and 7 (old)-month-old male and female *N. furzeri*. * *p* < 0.05, ** *p* < 0.01 vs. young females; ^$^
*p* < 0.05 vs. young males.

**Figure 4 ijms-24-03185-f004:**
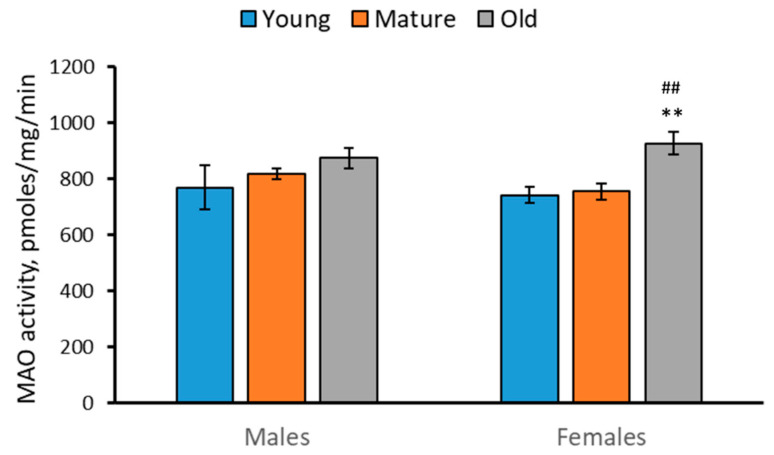
MAO activity in the brain of 2 (young)-, 4 (mature)- and 7 (old)-month-old males and females of *N. furzeri*. ** *p* < 0.01 vs. young females; ^##^
*p* < 0.01 vs. mature females.

**Figure 5 ijms-24-03185-f005:**
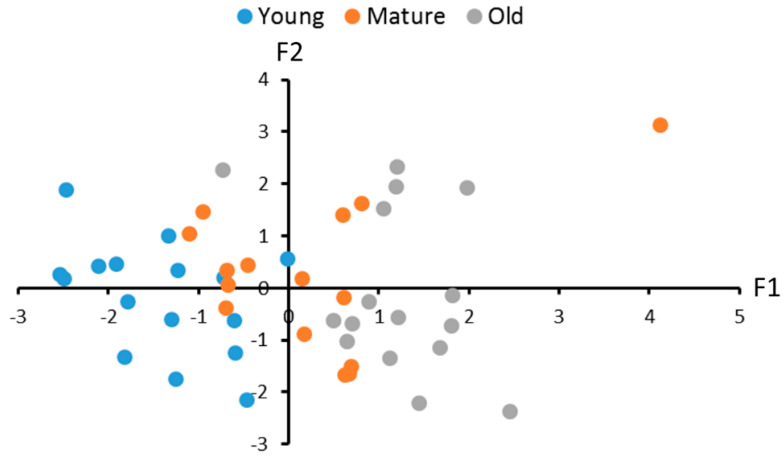
Individual scores for sixteen 2 (young)-month-old, fifteen 4 (mature)-month-old and sixteen 7 (old)-month-old males and females (1:1) of *N. furzeri* along with factor 1 and factor 2 yielded by the principal component analysis. These factors generalize the 5-HT and 5-HIAA levels, 5-HIAA/5-HT turnover rate and TPH and MAO activities in the brain. The coordinates of the two axes represent the factor scores of individual animals. All 16 scores corresponding to 2-month-old, six out of 15 scores corresponding to 4-month-old and one out of 16 scores corresponding to 7-month-old killifish are located in the left half of the graphic. Nine out of 15 scores corresponding to 4-month-old and fifteen out of 16 scores corresponding to 7-month-old killifish are located in the right half of the graphic.

**Table 1 ijms-24-03185-t001:** Two-way ANOVA of the effects of “Age”, “Sex” factors and their interaction on the variability of the 5-HT and 5-HIAA levels and the 5-HIAA/5-HT turnover rate in the brain of *N. furzeri*.

Trait	Age	Sex	Interaction
5-HT	F(2,41) = 38.23, *p* < 0.001	F(1,41) = 2.77, *p* = 0.1	F(2,41) = 2.42, *p* = 0.1
5-HIAA	F(2,41) = 1.37, *p* = 0.27	F(1,41) < 1	F(2,41) = 1.72, *p* = 0.19
5-HIAA/5-HT	F(2,41) = 11.69, *p* < 0.001	F(1,41) = 2.84, *p* = 0.1	F(2,41) < 1

**Table 2 ijms-24-03185-t002:** Pearson’s correlation coefficients between the TPH, MAO activities, the 5-HT and 5-HIAA levels and the 5-HIAA/5-HT turnover rate in the brain of 2-, 4- and 7-month-old male and female *N. furzeri*.

	TPH	MAO	5-HT	5-HIAA
TPH				
MAO	−0.04			
5-HT	**0.59**	−0.18		
5-HIAA	**0.50**	0.31	0.13	
5-HIAA/5-HT	−0.22	0.24	**−0.65**	**0.55**

Statistically significant values are in bold. The correlation coefficients significances were corrected according to Bonferroni.

**Table 3 ijms-24-03185-t003:** Factor loadings for the 5-HT and 5-HIAA levels, the 5-HIAA/5-HT turnover rate and the TPH and MAO activities in the brain of 2-, 4- and 7-month-old male and female *N. furzeri*.

	Factor 1 (41.5%)	Factor 2 (35.2%)	Factor 3 (16.4%)
TPH	**−0.61**	**0.68**	−0.11
MAO	0.40	0.42	**0.81**
5-HT	**−0.90**	0.26	0.10
5-HIAA	0.16	**0.95**	−0.17
5-HIAA/5-HT	**0.84**	0.40	−0.33

Statistically significant values are in bold. Factor loadings were calculated as the correlation coefficients between each original variable and this factor. Their significances were corrected according to Bonferroni.

## Data Availability

Not applicable.

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
