# Peer review of "Age-Related Alterations in the Level and Metabolism of Serotonin in the Brain of Males and Females of Annual Turquoise Killifish (Nothobranchius furzeri)"

_ijms, 2023, doi:10.3390/ijms24043185_

Round 1
Reviewer 1 Report
The manuscript ijms-2071369 entitled “Age-related alterations in the level and metabolism of serotonin in the brain of males and females of annual turquoise killifish (Nothobranchius furzeri)”investigated the level of of serotonin, its main metabolites in the brains of 2-, 4- and 7-month-old males and females of N. furzeri. In general, the study is interesting and the results can provide some additional information. However, the experimental design is very basic and the investigated parameters are very few and limited. Moreover, the results are not deeply discussed is the "discussion part" and the "discussion part" is mostly just the repeating the "results".
Author Response
Answer. In this study, we measured only age-related changes in the 5-HT, 5-HIAA levels, TPH and MAO activities in the brain of killifish males and females, since age-related alterations in the behavior and 5-HT-related genes expression have been already investigated in our earlier paper [44]. Taken together our present and earlier results provide considerable information concerning the changes in the brain 5-HT system in killifish during aging. We considerably corrected the Discussion and Conclusion, decreased the repeating the “results” and more deeply discussed the result impact. We thank the reviewer very much for his/her valuable comments.
Reviewer 2 Report
In submitted manuscript the age- dependent alterations in the brain 5-HT system of the short-lived turquoise killifish, Notobranchius furzeri have been examined
The authors studied the effect of age on the body mass, the level of serotonin, its main metabolites, 5-HIAA, as well as the activity of the key enzyme of its synthesis, tryptophan hydroxylases (TPH), and degradation, monoamine oxidase (MAO), in the brain of killifish, N. furzeri. Marked effect of age on the body mass, the level of serotonin, as well as the activity of TPH and MAO in the brain of killifish were revealed. The authors found that the level of 5-HT decreased, while 5HIAA/5-HT turnover increased with the age. The age related dynamics of TPH and MAO activities were related to the age related of Tph2 and Mao genes expression. The observed age-dependent alterations in 5-HT level, 5HIAA/5-HT turnover rate, TPH and MAO activities in brain of N. furzeri reflect the fundamental changes in the vertebrate 5-HT system during aging.
This work is the first study of age-depended alteration in the level and metabolism of serotonin in the brain of short-lived turquoise killifish. The authors have shown that the age-dependent alterations in the brain 5-HT system in N. furzer can model the aging changing in the brain of laboratory rodents, monkey and human
The results obtained by authors showed that N. furzeri is a perspective model to study the age-related alterations in the brain 5-HT system.
The manuscript contains new original data that are of great importance for the study of the fundamental problems of age-related alterations in the brain 5-HT system.
Presented work is written in clear style, broadly covers key literature and represents a new advancement in neuroscience. The manuscript is carefully prepared and contains very good documentation. In my opinion the paper by Evsiukova1 et al. can be published in International Journal of Molecular Sciences in the presented form
Author Response
Answer. We thank the reviewer very much for his/her appreciation of our research.
Reviewer 3 Report
1. Well done for the good job.
2. Minor English revision required, as the use of personal pronouns such as "we", "I", "me"... is now discouraged in scientific writing.
Concerns:
1. Why do the authors stick to one biomarker "serotonin" of ageing, why the choice of serotonin?
2. Biomarkers of longevity should have been checked as well.
3. Ageing in most cases is always accompanied by dementia and other neurological deficits, but the authors are restricted to only chemical biomarkers. Kindly justify.
4. There are some few observations in the attached file.

Author Response
- Well done for the good job.
- Minor English revision required, as the use of personal pronouns such as "we", "I", "me"... is now discouraged in scientific writing.
Answer. All phrases with personal pronouns were replaced by corresponding neutral phrases. English was corrected.
Concerns:
- Why do the authors stick to one biomarker "serotonin" of ageing, why the choice of serotonin?
Answer. Here we used the brain 5-HT system as the main object rather than “biomarker”. There are two main reasons for this choice: 1) the 5-HT system regulates numerous physiological functions, kinds of behavior, neuronal plasticity and it is involved in age-related neurodegeneration (see the Introduction, Discussion and Conclusion), 2) the 5-HT system is the main domain of our research.
- Biomarkers of longevity should have been checked as well.
Answer. Here we used two biomarkers of aging: 1) age-related morphological alterations such as reduced coloration in males, emaciation, spinal curvature, spine and face malformations and 2) sharp increase of deaths (lines 236-240).
- Ageing in most cases is always accompanied by dementia and other neurological deficits, but the authors are restricted to only chemical biomarkers. Kindly justify.
Answer. Learning and neurological deficits in aged killifish are well studied by other authors (Terzibazi et al., Exp. Gerant., 2007, 42:81-89; Matsui et al., Cell Reports, 2019, 26:1727-1733; Borgonova et al., Front. Neuroanat., 2021, 15:728720; Bagnoli et al., Aging Cell, 2022, 21:e13689), while our study the neurochemical alterations in the brain 5-HT system in killifish during aging is absolutely original.
- There are some few observations in the attached file.
Answer: In this study, we measured only age-related changes in the 5-HT, 5-HIAA levels, TPH and MAO activities in the brain of killifish males and females, since age-related alterations in the behavior and 5-HT-related genes expression have been already investigated in our earlier paper [44].
We thank the reviewer very much for his/her valuable comments and appreciation of our research.
Reviewer 4 Report
In this article entitled “Age-related alterations in the level and metabolism of serotonin in the brain of males and females of annual turquoise killi-fish (Nothobranchius furzer)” by Valentina S. Evsiukova et al. have studied the levels of serotonin and its main metabolites,5-hydroxyindoleacetic acid, as well as the activity of its key synthesis enzyme; tryptophan hydroxylases and monoamine oxidase enzyme responsible for 5-HT breakdown in the brain in 2-, 4- and 7-month-old males and females of N. furzeri. The authors showed a significant decrease in tryptophan hydroxylase activity and an increase in monoamine oxidase activity in the brain of 7-month-old females compared to 2-month-old females.
the noteworthy concerns I have are:
In terms of form, you have to respect the classical order of the redaction from of a scientific article: Introduction, Materials and Methods, Statistics, Results, Discussion, Conclusions and references.
Here, I provided additional comments per section:
Material and methods
the authors should provide details on the salinity rate of the sea water used in the study.
The date of publication of the protocol used in the “4.1animals” chapter has a slight inaccuracy that needs to be fixed (2022-2023).
The conclusion part of the manuscript should be shortened
CONCLUSIONS
what is the impact of this research paper results and their importance for humans?
Author Response
the noteworthy concerns I have are:
In terms of form, you have to respect the classical order of the redaction from of a scientific article: Introduction, Materials and Methods, Statistics, Results, Discussion, Conclusions and references.
Answer. The order Introduction, Results, Discussion, Materials and Methods, Statistics, Conclusions and References is one of the requirements of International Journal of Molecular Sciences and we may not change it.
Here, I provided additional comments per section:
Material and methods
the authors should provide details on the salinity rate of the sea water used in the study.
Answer. The information concerning water has been added (lines 226-227).
The date of publication of the protocol used in the “4.1animals” chapter has a slight inaccuracy that needs to be fixed (2022-2023).
Answer/ The number FWNR-2022-0023 is correct: the last four digits are the project individual number rather than year.
The conclusion part of the manuscript should be shortened
Answer: The Conclusion has been shortened.
CONCLUSIONS
what is the impact of this research paper results and their importance for humans?
Answer. Phrase concerning a translational impact of the present study has been added to the Conclusion end (lines 320-323). Two new references have been added [56,57].
We thank the reviewer very much for his/her valuable comments and appreciation of our research.
Reviewer 5 Report
The manuscript entitled „Age-related alterations in the level and metabolism of serotonin in the brain of males and females of annual turquoise killifish (Nothobranchius furzeri)“ describes age- and sex-related changes in metabolism of serotonin in brain of turquoise killifish. Study of turquoise killifish is important for medicine as recently available models in neuroscienses are time consuming.
Introduction in the manuscript is well written and contains information necessary for justification of undertaken study. Used material and methods are well described. The obtained results are clearly presented. Discussion is well written.
Some minor remarks:
line 89 instead of (Table.1) should be (Table 1).
line 132 - furzeri..
line 224 - gene..
line 248 - „into cold water (+2 C)“,
lines 256, 260, 293 and in other places: (+40C).
inconsistency in using uL (line 279) and ul (lines 283, 291 and other places)
line 310 - discrim-nant
Author Response
Answer. All these errors have been corrected. We thank the reviewer very much for his/her valuable comments and appreciation of our research
Round 2
Reviewer 1 Report
The experimental design is very simple and the investigated parameter are very limited. The manuscript dosent have enough content and scientific sound.
Author Response
Еру experimental design is fully consistent with the main aim of this study - the age-related alterations in the 5-HT, 5-HIAA levels as well in the TPH and MAO activities in the brain of males and females of N. furzeri.
Since, the age-related changes in the behavior and the 5-HT related genes expretion was published (Evsiukova et al., 2021) the set of parameters was limited by the 5-HT, 5-HIAA levels as well in the TPH and MAO activities.
Our results are new and original and we think that they have a considerable impact for understanding the alterations in the brain 5-HT systhem during aging.